# A Calibration Curve Implanted Enzyme-Linked Immunosorbent Assay for Simultaneously Quantitative Determination of Multiplex Mycotoxins in Cereal Samples, Soybean and Peanut

**DOI:** 10.3390/toxins12110718

**Published:** 2020-11-13

**Authors:** Yuxiang Wu, Jinzhi Yu, Feng Li, Jianlin Li, Zhiqiang Shen

**Affiliations:** 1School of Food Science and Pharmaceutical Engineering, Nanjing Normal University, Nanjing 210023, China; wuyuxiang@lvdu.net; 2Shandong Lvdu Biotechnology Co. LTD, Binzhou 256600, China; yujinzhi@lvdu.net (J.Y.); lifeng@lvdu.net (F.L.)

**Keywords:** calibration curve implanted ELISA, multiplex mycotoxins, high throughput detection, C plus plus programming language, analysis software

## Abstract

In this study, a rapid and sensitive immunoassay method has been established based on calibration curve implanted enzyme-linked immunosorbent assay (C-ELISA) for the simultaneously quantitative determination of aflatoxin B_1_, deoxynivalenol and zearalenone in cereal samples, soybean and peanut. The C-ELISA avoids using the standard substances during the detection. The principle of the C-ELISA is to implant the optimized standard curve data into the matched analysis software which can make data processing more convenient and faster. The implanted calibration curve software was programmed with C plus plus. In the new immunoassay system for aflatoxin B_1_, deoxynivalenol and zearalenone, their linear detection ranges were from 0.03~0.81, 1.00~27.00 and 5.00~135.00 ng/g, respectively. Recovery rates from spiked samples ranged from 85% to 110% with the intra-assay coefficients of variation under 5%. Compared with HPLC method, the new method showed consistence in all the observed contents of the three mycotoxins in real samples. The new method can rapidly and reliably high throughput simultaneously screen for multiplex mycotoxins.

## 1. Introduction

Mycotoxins are toxic secondary metabolites produced, inter alia, by fungi like *Aspergillus* and *Fusarium*, which often contaminate agricultural products, food, feed and the environment. Studies have shown that many mycotoxins are extremely toxic, teratogenic, mutagenic or carcinogenic [1,2,3]. The most common mycotoxins contaminated in grains are aflatoxin B_1_ (AFB_1_), deoxynivalenol (DON) and zearalenone (ZEN). Due to the serious health risks caused by mycotoxins, many international organizations and countries, including European Union (EU) and World Health Organization (WHO), have set maximum level residue for mycotoxins in food products at ppb level (μg/kg) [4,5]. For example, EU has established maximum permitted levels for 2 μg/kg AFB_1_, 1000 μg/kg DON and 75 μg/kg ZEN in cereal flour including maize flour, maize grits and maize flour [4]. Since most of the mycotoxins are chemically stable molecules even when cooked at quite high temperature, they can remain active for long time, which pose a threat to the health of human and farm animals [3,6].

Multiplex mycotoxins often simultaneously appear in a single cereal sample or soybean or peanut sample and could cause increased toxicity by additive and synergistic effects [7,8,9]. If assay for them is performed one by one, the assay efficiency would be low. Therefore, the development of a sensitive, rapid and good repeatability assay method for the simultaneous screening of multiplex mycotoxins is an urgent requirement for preventing mycotoxins from contaminating food chains.

Currently, chromatographic techniques combined with mass spectrometry such as high-performance liquid (HPLC-MS/MS) [10,11,12], gas chromatography-MS [13], microarray techniques [2,14,15], lateral flow immunoassay methods [16,17,18,19], electrochemical biosensors [20] and microplate [21] were used to simultaneously detect multiplex mycotoxins. However, chromatographic techniques are time-consuming, and they require expensive instruments and highly skilled personnel. Although microarray techniques are high throughput detection techniques for multiplex mycotoxins based on immunochemical principles, these detection systems often need manipulator sample spot which is expensive and difficult to be widely used for on-site screening mycotoxin assay [3]. Immunoassay methods for multiplex mycotoxins are simple, cheap and rapid. However, it is limited by the numbers of simultaneous assay target molecules. More importantly, the above assay methods for mycotoxins require mycotoxin standard substances, which causes operators to be easily exposed and in contact with the toxic substances.

In this work, we established a new multiplex mycotoxin assay system based on calibration curve implanted enzyme-linked immunosorbent assay (C-ELISA) with 96 well microplate for the simultaneously quantitative determination of AFB_1_, DON and ZEN in real samples. Our aim was to establish a new multiplex mycotoxin assay method without preparation of calibration curves to improve the sensitivity, accuracy and capacity of data processing. The implanted calibration curves were optimized and the implanted analysis software was designed by programming language/C plus plus (C^++^). The user only requires the optical density (OD) values of the different samples to be detected and the obtained data will be dealt with implanted calibration curves by the analysis software, which makes data processing more convenient and faster. Above all, the developed method does not require the mycotoxin standard substances for users, which avoids the operators being exposed and in contact with the toxic substances. The developed method has a great application potential for high throughput screening for other target molecules.

## 2. Results and Discussion

### 2.1. The Principles of the Developed Method

In the new multiplex mycotoxin protocol, the multiplex mycotoxins in samples are simultaneously extracted. The artificial antigenes of AFB_1_, DON and ZEN are coated on the 96 well microplate in different positions. The C-ELISA is performed with 96 well microplate for the simultaneously quantitative determination of AFB_1_, DON and ZEN. The optical density (OD) values of the different samples are detected and data are dealt with the implanted calibration curves by the analysis software. The new method does not require mycotoxin standard substances for calibration curves and avoids the toxic standard substances being exposed to operators. The implanted standardization for calibration curves can reduce the errors coming from the different operators. In addition, the data processing is more convenient and faster.

### 2.2. C-ELISA Calibration Curve

Three kinds of mycotoxins including AFB_1_, DON and ZEN have been used as model for C-ELISA calibration curves. The optimal working concentration of mycotoxin artificial antigens and monoclonal antibodies for mycotoxins were optimized by the checkerboard method. The optimal coating concentrations of artificial antigen was 0.5 μg/g for AFB_1_, DON and ZEN, respectively. The working concentrations of monoclonal antibody were 0.1, 0.025, 0.0625 μg/g for AFB_1_, DON and ZEN, respectively. Under the optimal working concentrations of mycotoxin artificial antigens and monoclonal antibodies, the optimal mycotoxin calibration curve for the three mycotoxins is shown in Figure 1. The 50% inhibition concentration (IC_50_), limit of detection (LOD) values (signal/noise = 3) and the working ranges were shown in Table 1. The logistic correlation coefficients (R) were all above 0.950, indicating a good correlation between the concentration of the mycotoxins and inhibition rate. The IC_50_ values are 0.093, 3.38 and 17.70 ng/g for AFB_1_, ZEN and DON, respectively. LOD values are 0.03, 1.00 and 5.00 ng/g for AFB_1_, ZEN and DON, respectively. The linear detection ranges are 0.03~0.81, 1.00~27.00 and 5.00~135.00 ng/mL, for AFB_1_, ZEN and DON, respectively. The optimum calibration curve has been implanted in the software and used in the following experiments.

### 2.3. Optimization of Sample Pretreatment in C-ELISA Assay

We found that the composition of sample diluent seriously affects the results of C-ELISA. Sample diluents including PBS (pH = 7.4, 0.1 mol/mL), PB (pH = 7.4, 0.1 mol/mL) and Tris-HCL (0.1 mol/mL) have been investigated in the spiked samples. The coefficient of variation (CV), recovery rate, false positive rate and false negative rates were calculated and the results were shown in Table 2. For Tris-HCL (0.1 mol/mL) diluent, the recovery rates of the three mycotoxins are between 65 and 100%, with 0% false positive rate and 11–18% false negative rate. Under the sample dilution of pH 7.4, 0.1 mol/L PBS, the recovery rates are between 81 and 119%, with 1–2% false positive rate and 0% false negative rate for the three mycotoxins. Under the sample dilution of pH 7.4, 0.1 mol/L PB, the recovery rates are between 81 and 119%, with 6–9% false positive rate and 0% false negative rate. Therefore, pH = 7.4, 0.1 mol/L PBS is taken as the optimum sample diluent. The precision assays were carried out in five replicates for intra-assay and inter-assay evaluation. The standard deviations of intra-assay and inter-assay for the three mycotoxins are less than 5.0 and 6.0%, respectively. The reaction dynamics of antigens and antibodies are often influenced by pH and the ion strength of sample diluent, which would influence the OD values of samples. Therefore, the composition of sample diluent is a key factor for the C-ELISA Assay.

### 2.4. Specificity Evaluation

The specificity for the new developed method has been evaluated by IC_50_ values and cross-reaction ratios among the analogues or other mycotoxins and the target mycotoxin. The results are shown in Table 3. IC_50_ values for the target mycotoxins are the lowest among them. The cross-reaction ratios are 100% for AFB_1_, ZEN and DON and are less than 15% for analogues of AFB_1_ and ZEN and other mycotoxins. These results indicate that the new method has a good specificity for AFB_1_ and ZEN. However, for DON, the cross-reaction ratios are more than 70% for analogues of DON and less than 0.1% for other mycotoxins. The detection specificity for multiplex assay mainly depends on the specificities of antibodies to mycotoxins. Therefore, improving the specificities of antibodies to mycotoxins would increase the detection specificities of the new method.

### 2.5. Comparison with HPLC

The C-ELISA assay method has been further verified by the classical HPLC method for mycotoxin assay. The comparison between the new developed method and HPLC method was performed in 30 blind cereal samples including wheat (S2–S5), corn (S6–S15, S23), soybean (S1, S24–S30) and peanut (S16–S22) samples. The result is shown in Table 4. The results for S1, S3, S6, S7, S9~S14, S16~S22, S24~S28 and S30 show that they have no mycotoxins. The other samples show that the results of the new developed detection method are in agreement with that of HPLC method and the deviations of detection were all below 15%. The influences of the different sample substrates coming from wheat, corn, soybean and peanut on the results of assay have not been observed in the samples.

Chromatographic techniques, microarrays and lateral flow immunoassays are the most common methods for mycotoxin multiplex screening assay [22,23]. Both of the first two techniques require expensive instruments and highly skilled personnel, which limited their on-site application in practice [24]. Lateral flow immunoassays have been successfully developed to screen multiplex mycotoxin on-site assay [16,25]. However, most of them are qualitative or semiquantitative detection and the number of multiplex simultaneous screening mycotoxins is limited [26]. In addition, until now, most of quantitative assay methods including common ELISA require the mycotoxin standard substances for quantitative detection, which increases the safety risk of operators and cost of assay. This new C-ELISA method has been established in the work, which does not need the mycotoxin standard substances for assay. The implanted calibration curve software was programmed with C^++^ and the optimum conditions of application was established. The C-ELISA method has been verified to be reliable, sensitive and rapid for multiplex screening for mycotoxins in real samples.

## 3. Conclusions

In the work, a new rapid, sensitive, high throughput, cost-efficient method for multiplex mycotoxins has been established by the implanted calibration curve into the software. The optimum conditions of the implanted calibration curve have been investigated. The composition of sample diluent seriously affects the results of C-ELISA. The new developed method does not require the standard substances, which avoids the contact opportunity of toxic reagent for the operator. Using the new method, data processing is more convenient and faster. The detection results of blind samples show that there is no difference between the developed method and HPLC method. The established method has a great application potential for multiplex mycotoxin assay.

## 4. Experimental Section

### 4.1. Materials and Reagents

The cereal samples, soybean and peanut samples were provided by the Academy of State Administration of Grain (Beijing, China). The mycotoxin artificial antigens including AFB_1_-BSA (bovine serum albumin), ZEN-BSA, DON-BSA and their mouse monoclonal antibodies (mAbs) were purchased from Shandong Landu Biotechnology Co., Ltd., China). DON, AFB_1_, ZEN, aflatoxin B_2_ (AFB_2_), aflatoxin G_1_ (AFG_1_), aflatoxin G_2_ (AFG_2_), aflatoxin M_1_ (AFM_1_) aflatoxin M_2_ (AFM_2_), zearalenol (ZEA), zeranol (ZAA), 3-acetyldeoxynivalenol(3A-DON), 15-Acetoxyscirpenol(15-ADON), Trichothecenes (T2) standard substances were purchased from Fermentek (Jerusalem, Israel). Tris-base and 3,3’,5,5’-Tetramethylbenzidine (TMB) were purchased from NanJing SunShine Biotechnology Co., LTD (Nanjing, China). Bovine serum albumin (BSA) and goat anti-mouse secondary antibody-horseradish peroxidase (HRP) were bought from Sigma-Aldrich (St. Louis, MO, USA). Tween-20, methanol and sulfuric acid were purchased from Sinopharm (Beijing, China).

### 4.2. Equipment

The following equipment were used in the work: microplate reader (MK3, Thermo, Waltham, MA, USA), Agilent 1290 Infinity LC (Santa Clara, CA, USA), incubator (DRP-9272, Senxin, Shanghai, China), high-speed centrifuge (HettichRotofix46, Hettich, Germany) and solid phase extraction column (Oasis HLB, Milford, CT, USA).

### 4.3. Optimization of C-ELISA Standard Curve

100 μL of different concentrations (0.1, 0.2, 0.5 and 1 μg/g) of mycotoxin artificial antigens were respectively coated in separate wells of 96-microplate. After each well was washed with 0.01 mol/L pH 7.4 Tween-20 (0.05%)-phosphate buffer solution (PBST) for three times and blocked by 100 μL of 3% BSA. The microplate was incubated at 4 °C for 24 h and then washed with PBST for three times. 50 μL of different concentrations (0.025, 0.05, 0.1, and 0.2 μg/g for AFB_1_, 0.0125, 0.025, 0.05 and 0.1 μg/g for DON and 0.01, 0.03, 0.0625, 0.125 and 0.25 μg/g for ZEN, respectively) of mouse monoclonal antibodies of mycotoxins and 50 μL of a certain concentration of mycotoxin standard substances (0, 0.03, 0.09, 0.27 and 0.81 ng/g for AFB1, 0, 1, 3, 9 and 27 ng/g for ZEN, 0, 5, 15, 45 and 135 ng/g for DON, respectively) were respectively added in 96-well microplate and incubated at 37 °C for 1 h, then washed with PBST for three times. A certain concentration of HRP-goat anti-mouse secondary antibody was added in the each well and cultivated at 37 °C for 1 h, then washed with PBST for three times. 0.1mLof TMB substrate solution was added in each well. After incubation at room temperature in the dark for 10 min, the reaction was terminated by adding 0.1mL of 1 M H_2_SO_4_. In each plate well the absorbance at 450 nm was determined in the ELISA reader. The checkerboard method was used to determine the working concentrations of the coated antigen and antibody. The absorbance (OD450) of the blank control at 450 nm was 1.5–2.0.

The inhibition calibration curves were drawn by plotting the OD value (sample)/the OD value (blank control) against the logarithm of the analyte concentration using the Origin 9.3 software (OriginLab Corp., 9.3). The calibration curves were obtained and implanted in the designed analysis software.

### 4.4. The Designed Analysis Software

The analysis software implanted with calibration curves was designed by C^++^ and run in Windows XP or higher Window vision work environment. The data system is SQLite. The interface of the designed analysis software was shown in Figure 2. The data was processed by inputting Std0 (blank control). Here, N, V and F are the sample name, the OD value and the sample dilution factor, respectively. The data was analyzed by inputting the data of blank control and the corresponding sample parameters.

### 4.5. Sample Preparation for C-ELISA Assay

The real sample preparation is treated according to the reference [2]. In detail, the samples including wheat, corn, soybean and peanut were respectively ground by a high-speed disintegrator and particles were less than 1mm. 5 g of the ground sample was added into a 100 mL flask. A series of the different mycotoxin standard substances in 100% methanol were respectively added in the flask and completely mixed. The spiked samples were respectively extracted with 20 mL of methanol-water (4:1 *v/v*) at 10× g shaking for 5 min and then the extracts were centrifuged at 3000× g for 5 min. 100 μL of supernatant solution was separated and diluted with diluent(PBS) at a ratio of 1:4.

### 4.6. Detection of Samples by C-ELISA

50 μL of different extraction solutions of samples, 50 μL of mouse monoclonal antibodies of mycotoxins and 50 μL of HRP-goat anti-mouse secondary antibody were respectively added into 96-well microplate coated with the different artificial antigens and incubated at 37 °C for 1h. The microplate was washed with PBST for three times. 0.1mLof TMB substrate solution was added in each well. After incubation at room temperature in the dark for 10 min, the reaction was terminated by adding 0.1mL of 1 M H_2_SO_4_. In each plate well the absorbance at 450 nm was determined in the ELISA reader. The OD values were put in the software of C-ELISA and the concentrations of mycotoxins in samples were obtained according to the implanted calibration curves.

### 4.7. Specificity Evaluation

The specificity evaluation of the developed method has been performed by the comparison of IC_50_ and cross-reaction ratios among the analogues (AFB_2_, AFG_1_, AFG_2_, AFM_1_ and AFM_2_ for AFB_1_, 3A-DON and 15-ADON for DON, ZEA and ZAA for ZEN, respectively) or other mycotoxins (OTA and T2) and the target mycotoxin. The procedures for calibration curves of the analogues and other mycotoxins were obtained by the same as that in 2.3 method except the analogues and other mycotoxins as target analytes. The cross-reaction ratios were calculated according to the following equation:
S = y/Z × 100%
(1)

Here, S is cross-reaction ratio. y is the antigen concentration corresponding to their IC_50_ and Z is the concentration of the analogues or other mycotoxins corresponding to their IC_50_.

### 4.8. Detection of Samples by HPLC

The HPLC detections for mycotoxins for the spiked and real samples were carried out on Agilent 1290 Infinity LC (USA), as well as this C-ELISA. In detail, the conditions for ZEN are as follows: 1.0 mL/min flow rate, C_18_Agilent XDB 4.6 × 250 mm, column temperature at room temperature, 100 μL of injection volume, acetonitrile: water solution: methanol (46:46:8, *v/v*/*v*) as mobile phase, fluorescence detector at 274 nm for excitation and 440 nm emission wavelength, respectively. Before analysis for AFB_1_, the sample was derivatized by 300 μL of trifluoroacetic acid for 15 min at 40 °C. After that, the sample was dried by nitrogen at 50 °C water bath and dissolved into 200 μL of mixture solution (1:1 methanol: acetonitrile, *v/v*). The sample was filtered through a filter membrane with pore size of 220 nm. The filtrate was collected and induced into HPLC apparatus. HPLC analysis conditions were as follows: C_18_Agilent XDB 4.6 × 250 mm, 1.0 mL/min flow rate, column temperature at 30 °C, methanol: water (45: 55, *v/v*) as mobile phase, fluorescence detector at 360 nm for excitation and 440 nm emission wavelength, respectively. The conditions for DON are as follows: C_18_Agilent XDB 4.6 × 250 mm, 20 μL of injection volume, 0.8 mL/min flow rate, column temperature at 35 °C, methanol: water (20: 80, *v/v*) as mobile phase, UV-detector at 218 nm. The assay was performed five times for each sample. The averages calculated from the results in the two methods were compared to verify the accuracy and reliability.

### 4.9. Data Analysis

These data were dealt with Origin9.3 (OriginLab Corp., Northampton, MA, USA). The test was repeated at least 3 times to statistically calculate the average value and standard deviation.

## Figures and Tables

**Figure 1 toxins-12-00718-f001:**
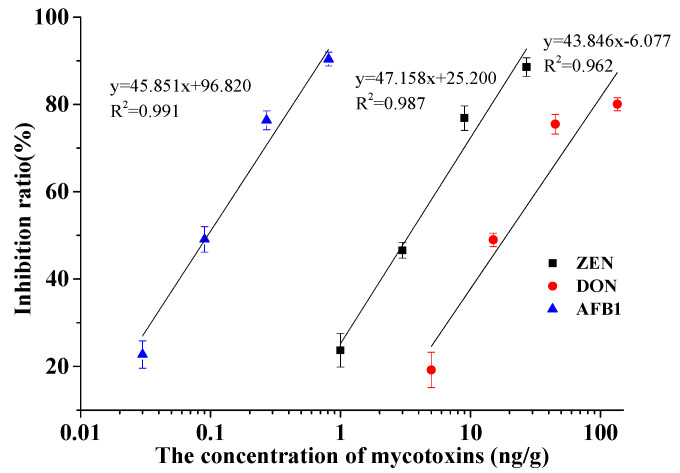
The optimal mycotoxin calibration curves for implanting in the analysis software.

**Figure 2 toxins-12-00718-f002:**
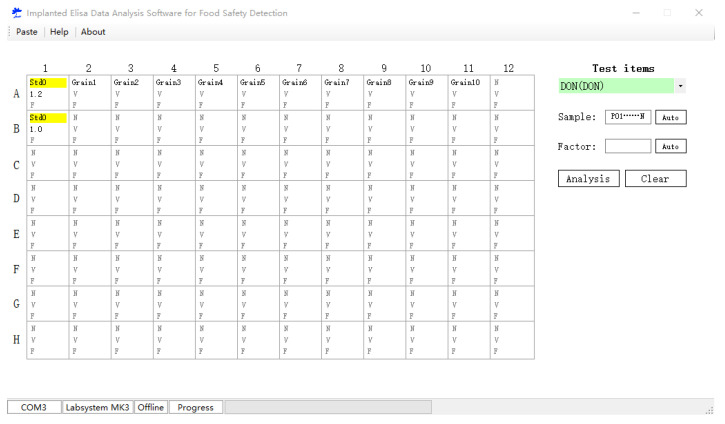
The interface of the analysis software implanted with calibration curves.

**Table 1 toxins-12-00718-t001:** The optimal mycotoxin calibration curves for the three mycotoxins in the detection assay.

Analytes	Standard Curve	IC_50_ (ng/g)	LOD/(ng/g)	Working Range/ (ng/g)
AFB_1_	y_AFB1_ = −45.851x + 96.820, R^2^ = 0.991	0.093	0.03	0.03~0.81
ZEN	y _ZEN_ = −47.158x + 25.200, R^2^ = 0.987	3.38	1.00	1.00~27.00
DON	y_DON_ = −43.846x − 6.077, R^2^ = 0.962	17.70	5.00	5.00~135.00

**Table 2 toxins-12-00718-t002:** The results measured by the C-ELISA in spiked cereal samples.

Analytes	Spiked Concentration (ng/g)	CV (%)	0.1 mol/L Tris-HCL			0.1 mol/L PBS			0.1 mol/L PB		
			Recovery (%)	False Positive Rate (%)	False Negative Rate (%)	Recovery (%)	False Positive Rate (%)	False Negative Rate (%)	Recovery (%)	False Positive Rate (%)	False Negative Rate (%)
**ZEN**	100	0.9~1.1	65~95	0	11	85~115	1	0	83~116	6	0
	300		68~97			85~114			85~115		
	600		65~99			81~119			81~119		
**DON**	200	1.3~1.7	68~96	0	15	85~116	2	0	81~115	9	0
	1000		67~99			84~115			82~116		
	2000		65~99			81~119			85~119		
**AFB_1_**	1	5.1~5.2	63~93	0	18	85~119	1	0	85~115	7	0
	5		68~99			81~115			85~119		
	20		66~100			85~115			81~114		

**Table 3 toxins-12-00718-t003:** The specificity evaluation.

Analytes	IC_50_ (ng/g)	S	Analytes	IC_50_ (ng/g)	S	Analytes	IC_50_ (ng/g)	S
AFB_1_	0.093	100	ZEN	3.38	100	DON	18.89	100
AFB_2_	0.76	12.2	ZEA	27.7	12.2	3A-DON	20.67	91.4
AFG_1_	0.87	10.7	ZAA	42.3	7.9	15-ADON	23.65	79.9
AFG_2_	0.98	9.5	DON	>5000	<0.1	AFB_1_	>10000	<0.1
AFM_1_	1.75	5.3	T2	>5000	<0.1	AFM_1_	>10000	<0.1
AFM_2_	1.97	4.7	OTA	>5000	<0.1	AFG_1_	>10000	<0.1
ZEN	>1000	<0.01	AFB1	>5000	<0.1	ZEN	>10000	<0.1
DON	>1000	<0.01	AFM1	>5000	<0.1	T2	>10000	<0.1
OTA	>1000	<0.01	AFG1	>5000	<0.1	OTA	>10000	<0.1

**Table 4 toxins-12-00718-t004:** Comparison between C-ELISA and HPLC.

Sample	C-ELISA (μg/kg)			HPLC(μg/kg)		
	ZEN	DON	AFB_1_	ZEN	DON	AFB_1_
S2	ND	ND	5.6 ± 0.07	ND	ND	6.2 ± 0.02
S4	423.2 ± 0.08	ND	ND	412.2 ± 0.19	ND	ND
S5	ND	956.2 ± 0.11	ND	ND	935.3 ± 0.25	ND
S8	263.2 ± 0.12	ND	ND	265.2 ± 0.13	ND	ND
S15	ND	ND	4.6 ± 0.03	ND	ND	4.2 ± 0.06
S23	412.3 ± 0.16	ND	ND	423.2 ± 0.16	ND	ND
S29	ND	ND	3.2 ± 0.08	ND	ND	3.6 ± 0.07
S1, S3, S6, S7, S9~S14, S16~S22 S24~S28, S30	ND	ND	ND	ND	ND	ND

Each was determined with 3 repeats. ND not detectable.

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
