# Peer review of "A Calibration Curve Implanted Enzyme-Linked Immunosorbent Assay for Simultaneously Quantitative Determination of Multiplex Mycotoxins in Cereal Samples, Soybean and Peanut"

_toxins, 2020, doi:10.3390/toxins12110718_

Round 1
Reviewer 1 Report
The work has been improved.
In my opinion it can be accepted in the present form.
Author Response
The English language and style have been carefully revised.

Reviewer 2 Report
The comments are attached in the file.

Author Response
Manuscript Number: toxins-981351
Title: “A Calibration Curve Implanted Enzyme-Linked Immunosorbent Assay for Simultaneously Quantitative Determination of Multiplex Mycotoxins in Cereal Samples, Soybean and Peanut”
Review
This report proposes a variant of the ELISA method to optimize the calibration curves for the detection and simultaneous quantification of three mycotoxins, i.e.: aflatoxin B1, deoxynivalenol and zearalenone in samples of cereals, soybeans and peanuts. The applied procedure assumes that the prepared calibration curves allow to determine target mycotoxins without the use of standard mycotoxin solutions. The paper presents mainly a method of the calibration curves preparation (as it was properly described in the title of the manuscript). The mentioned curves were tested to determine simultaneously several mycotoxins by its implementation in tools mentioned in the methodology. The principles of these tools operation was not described. Despite the fact that the approach discussed in the paper is interesting and has an application value, the description of the methodology at some points is too general, which would make it difficult to reproduce the prepared calibration curves and their further use. In my opinion the submitted manuscript is a valuable contribution to the improvement of the existing diagnosing techniques, but there are some comments and suggestions that should be explained and extended in order to improve the quality of the manuscript:
Response:
We partially agree. The principles of these tools operation have been added in 3.1 section. The calibration curves are given to user and does not require to prepare the calibration curves, which is better than the calibration curves prepared by user. The calibration curves can be easily reproduced and further use.
Introduction
- 1, l. 22-23: The first sentence of this section: “… metabolites produced by fungi like aspergillus and fusarium, which often contaminate agricultural products, food, feed and the environment…”
should be specified as follows "…metabolites produced, inter alia, by ...”, as there are also other species that produce mycotoxins, e.g. ochratoxin A (OTA), which is the second most important and naturally occurring mycotoxin after aflatoxin produced by some kinds of Aspergillus and species of Penicillium. It is mainly produced in cereals by moulds: Aspergillus ochraceus, A. carbonarius and Penicillium verrucosum. Moreover, the names of species should start with a capital letter.
Response:
We agree. The first sentence of this section has been revised: Mycotoxins are toxic secondary metabolites produced, inter alia, by fungi like Aspergillus and Fusarium, which often contaminate agricultural products, food, feed and the environment.
- 2, l. 50-59: The text has a structure of a conclusion or an abstract and refers to the realized points of the work, whilst in this part of manuscript the authors should indicate what the aim of the study was.
The study entitled: “Rapid Multiple Immunoenzyme Assay of Mycotoxins “ by Urusov et al. which deals with similar topic, and which was previously published in Toxins 2015, 7, 238-254; doi:10.3390/toxins7020238 should be mentioned in the introduction and discussion.
Response:
We agree. Our aim was to establish a new multiplex mycotoxin assay method without preparation of calibration curves to improve the sensitivity, accuracy and data processing capacity. The aim has been added in line 53-55.
The reference has been cited in the manuscript.
2. Experimental Section
These parts are described a bit superficially.
- 2, l. 78-79: Comment to the sentence: “100 μL of different concentrations of mycotoxin artificial antigens (5mg/mL) were respectively coated in 96-well microplate.”
As it is part of methodology, it is advisable to provide specific names of the used mycotoxin artificial antigens and their concentrations applied in the experiments. What does the given value of “5mg/mL” apply to, if the different concentrations of mycotoxin artificial antigens were used? Moreover, it should be clearly write that the different artificial antigens were incubated “in separate wells of microplate”.
Response:
We agree.
The “0.1, 0.2, 0.5,1 μg/g” artificial antigens were incubated “in separate wells of microplate” has been added in manuscript.
5mg/mL is origin concentration of artificial antigens.
- 2, l. 81-83: The same comment is related to the sentence: “50 μL of different concentrations of mouse monoclonal antibodies of mycotoxins (5mg/mL) and 50 μL of a certain concentration of mycotoxin standard substance were respectively added in 96-well microplate…”
Response:
We agree.
The sentences have been revised: 50 μL of different concentrations (0.025, 0.05, 0.1, and 0.2 μg/g for AFB1, 0.0125, 0.025, 0.05, and 0.1 μg/g for DON and 0.01, 0.03, 0.0625, 0.125,and 0.25 μg/g for ZEN) of mouse monoclonal antibodies of mycotoxins and 50 μL of a certain concentration of mycotoxin standard substances (0,0.03,0.09,0.27,and 0.81ng/g for AFB1,0,1,3,9, and 27 ng/g for ZEN, 0,5,15,45,and 135 ng/g for DON, respectively )were respectively added in 96-well microplate and incubated at 37°C for 1 h, then washed with PBST for three times.
- 3, l. 110; p. 4, l. 123: the names of the used diluents and the name used analogues and other mycotoxins should be mentioned in experimental procedure.
Response:
We agree.
The analogues (AFB2, AFG1, AFG2, AFM1 and AFM2 for AFB1, 3A-DON and 15-ADON for DON, ZEA and ZAA for ZEN, respectively) or other mycotoxins (OTA and T2) have been added.
- 2, l. 81: It would be better do give the exact incubation time expressed in hours (h) instead of the term “overnight”.
Response:
We agree. 24 h has been added.
- 2, l. 84, 85 and p. 3, l. 114: How can the differences in incubation time be explained for mouse monoclonal antibodies of mycotoxins, HRP-goat anti-mouse secondary antibody, mycotoxins and extraction solutions of samples in procedures described in 2.3. Optimization of C-ELISA Standard Curve and 2.6. Detection of Samples by C-ELISA subsections?
Response:
We agree. This is an error in 2.6. The incubation time has been revised.
- 3, l. 95-100: The authors indicated the tools used in the study to implement and test the calibration curves. It is suggested to describe how the program operates. Does the program do more than read data from the three calibration curves?
Response:
We agree. The program operation has been described in section 2.4. Yes, more than thousands of data could be read.
3. Results and Discussion
- 5, l. 159: It is not clear what does it mean “N” in the equation “(S/N=3)”, as the symbol “N” in the subsection 2.4. The Designed Analysis Software was used as the sample name.
Response:
We agree. “N” in the equation “(S/N=3)” in line 159 in page5 has been revised as “signal/noise =3”. “N” in the subsection 2.4 means number.
- 5, l. 161-165: The text: ”The IC50 values are 0.093, 3.38 and 17.70 ng/g for AFB1, ZEN and DON, respectively. LOD values are 0.03, 1.00 and 5.00 ng/g for AFB1, ZEN and DON, respectively. The linear detection ranges are 163 0.03~0.81,1.00~27.00 and 5.00~135.00 ng/mL, for AFB1, ZEN and DON, respectively.” is unnecessary, as values included in the text are repetition of the same values presented in the Table 1. Moreover, the value of the IC50 for DON given in the text (“17.70 ng/g” - p. 5, l. 162)differs from that given in the Table 1.
Response:
We partially agree. The data in the text explained the Table 1. IC50 for DON given in the Table 1 has been revised.
In the study, the grain samples were not naturally contaminated, but spiked with mycotoxins. It is difficult to say whether the method would also give satisfactory results on natural cereal, soybean and peanut samples. Therefore, further tests with the use of naturally contaminated samples are advisable.
Response:
We partially agree. The natural contaminated cereal samples have been researched in the section 3.5. The spiked samples have been researched in Table2.
